# Kinematic Evaluation of the Sagittal Posture during Walking in Healthy Subjects by 3D Motion Analysis Using DB-Total Protocol

**DOI:** 10.3390/jfmk7030057

**Published:** 2022-08-11

**Authors:** Paolo De Blasiis, Allegra Fullin, Mario Sansone, Angelica Perna, Silvio Caravelli, Massimiliano Mosca, Antonio De Luca, Angela Lucariello

**Affiliations:** 1Section of Human Anatomy, Department of Mental and Physical Health and Preventive Medicine, University of Campania “Luigi Vanvitelli”, Via Luciano Armanni 5, 80138 Naples, Italy; 2Department of Electrical Engineering and Information Technology (DIETI), University of Naples Federico II, Via Claudio 21, 80125 Naples, Italy; 3Department of Medicine and Health Sciences “Vincenzo Tiberio”, University of Molise, Via F. De Santis, 86100 Campobasso, Italy; 4II Clinic of Orthopaedics and Traumatology, IRCCS Istituto Ortopedico Rizzoli, 40136 Bologna, Italy; 5Department of Sport Sciences and Wellness, University of Naples “Parthenope”, 80100 Naples, Italy

**Keywords:** posture, spine, sagittal alignment, kinematics, 3D gait analysis, stereophotogrammetry

## Abstract

Posture can be evaluated by clinical and instrumental methods. Three-dimensional motion analysis is the gold standard for the static and dynamic postural assessment. Conventional stereophotogrammetric protocols are used to assess the posture of pelvis, hip, knee, ankle, trunk (considered as a single segment) and rarely head and upper limbs during walking. A few studies also analyzed the multi-segmental trunk and whole-body kinematics. Aim of our study was to evaluate the sagittal spine and the whole-body during walking in healthy subjects by 3D motion analysis using a new marker set. Fourteen healthy subjects were assessed by 3D-Stereophotogrammetry using the DB-Total protocol. Excursion Range, Absolute Excursion Range, Average, intra-subject Coefficient of Variation (CV) and inter-subject Standard Deviation Average (SD Average) of eighteen new kinematic parameters related to sagittal spine and whole-body posture were calculated. The analysis of the DB-Total parameters showed a high intra-subject (CV < 50%) and a high inter-subject (SD Average < 1) repeatability for the most of them. Kinematic curves and new additional values were reported. The present study introduced new postural values characterizing the sagittal spinal and whole-body alignment of healthy subjects during walking. DB-Total parameters may be useful for understanding multi-segmental body biomechanics and as a benchmark for pathological patterns.

## 1. Introduction

Posture is defined as the position of the human-body segments and their orientation in space [1]. Postural Control System (PCS) processes the peripheral sensory afferents and modulates the muscle chains in order to keep the projection of the center of body mass between the two feet against the force of gravity, ensuring minimal energy expenditure through functional neuromuscular adaptation and biomechanical strategies [1,2]. PCS acts continuously both in static and dynamic phases, guaranteeing balance in the most unstable postural conditions [3]; however, its precise functioning may be affected by age, cognitive–motor factors and organic pathologies [4], therefore an accurate postural analysis may be useful during medical diagnostic–therapeutic process. Posture can be evaluated by clinical and instrumental methods: while the former is greatly prone to inter-operator variability, the latter allows the quantification of any postural elements, such as postural stability [2], morphology and symmetry [5] of musculoskeletal districts and/or plantar pressures in static and dynamic conditions. The gold standard for the three-dimensional analysis of whole-body posture in standing and during various motor tasks is 3D stereophotogrammetry [6]. This technology uses an optoelectronic system consisting of cameras sending and receiving infrared rays reflected by skin markers placed on specific body landmarks of the subject to be evaluated; subsequently, a software processes the signals and reconstructs a three-dimensional image of the posture in relation to the marker set used. Traditional stereophotogrammetric models [7,8,9,10,11] were used to assess the kinematics of pelvis, hip, knee, ankle, trunk (considered as a single rigid segment that does not provide information on kinematic changes within the spine) and rarely head and upper limbs. A few studies analyzed the multi-segmental-trunk, head, lower- and upper- limbs kinematics in upright standing [12,13,14] by introducing a larger number of skin markers on the body; other ones assessed the trunk in several districts during locomotion or elementary exercises [15,16,17,18,19,20,21,22]. In particular, [15] described spine kinematics considering lower-thoracic, lumbar and pelvic segments during walking; [16] characterized spine motion by 5-link-segment-model to upright posture, chair raising-sitting, stepping up and down, and level walking, and [17] investigated the contribution of upper-body movements to dynamic balance control during different and challenging motor tasks. Other studies evaluated the kinematic mechanisms within the spine during walking using the following stereophotogrammetric parameters: Sagittal Vertical Axis (SVA), Pelvic Tilt (PT), and lordosis and kyphosis angles. The latter were assessed in young asymptomatic volunteers [18], in elderly females [19], in patients with adult spinal deformities (ASD) compared to controls without treatment [20] and after spinal surgery [21], showing changes in sagittal alignment and compensation strategies both before and after surgical correction. However, no study considered whole-body kinematic parameters as we performed in our recent study [14], introducing a new marker set titled DB-Total protocol, that placed additional skin markers on the head, spine and upper limbs with respect to the Helen Hayes marker set [11] for better evaluating the sagittal spine and whole-body posture of patients with late-onset Pompe disease compared to healthy controls in upright standing. Moreover, DB-Total protocol showed high reliability for the assessment of the sagittal spine thanks to strong correlations between the stereophotogrammetric and radiographic parameters. Therefore, aim of the present study was to analyze the DB-Total parameters in an adult healthy population during walking in order to define their kinematic curves, additional values (mean of Average, Excursion Range, Absolute Excursion Range), intra- and inter-subject repeatability.

## 2. Materials and Methods

### 2.1. Subject Population

A cross-sectional observational study was performed, recruiting fourteen healthy subjects (7 females and 7 males; mean ± SD: age = 46.7 ± 14.9 y (median = 45), height = 1.71 ± 0.10 m, weight = 72.3 ± 21.6 kg and BMI = 23.44 ± 2.99 kg/m^2^; reported in Table 1) at Functional Anatomy Laboratory of University of Campania “Luigi Vanvitelli”. The following inclusion criteria were used: age between 30 and 60 years old and normal weight. Exclusion criteria: presence of pain, muscle-skeletal injuries in the last 3 months, neurological and visual disease, postural and spinal disorders, previous orthopedic surgery and cognitive impairment.

### 2.2. Data Acquisition

Each subject underwent 3D stereo-photogrammetric examination that was performed using an optoelectronic system composed of eight Smart-D cameras (BTS Bioengineering, Milano, Italy) set at a frequency of 100 Hz and two force platforms (BTS Bioengineering, Milano, Italy). Reflective markers were placed on body landmarks according to the DB-Total marker set [14]. The latter was proposed to comprehensively investigate spinal and whole-body kinematics and extend the conventional Helen Hayes M.M. protocol [11], including 37 markers placed on the following body landmarks: nasion (Ns), frontozygomatic suture (FZs), spinous processes of C7-T7-T12-L3-L5-S2, acromioclavicular joint (ACj), epicondylus humeri (eH), ulnar styloid (Us), anterior superior iliac spine (ASIS), greater trochanter (gT), medial (mEF) and lateral (lEF) epicondylus femoris, fibular head (Fh), medial (mM) and lateral (lM) malleoli, I°–III° and V° metatarsal heads (MtH) and heel (He) bilaterally (as described in Figure 1A). These markers were always placed on body landmarks by the same expert operator (physiatrist, Ph.D., expert in functional anatomy and 3D motion analysis). Subsequently, each subject performed three consecutive trials in the same direction, walking barefoot on a 6 m walkway at a self-selected normal-pace speed.

### 2.3. Data Processing

Raw data were processed with Smart Analyzer software (BTS Bioengineering, Milano, Italy). Seven types of conventional spatial–temporal data (cycle duration, cadence, gait speed, stance phase, swing phase, double-support phase, stride length and step width) and conventional kinematic parameters of traditional marker-set protocols [11] were computed. In addition, the DB-Total protocol permitted us to calculate the following eighteen kinematic parameters: C7–Nasion Angle (CNA), T7–Nasion Angle (TNA), S2–Nasion Angle (SNA), Heel–S2 Angle (HSA), S2–C7 Angle (SCA), S2–T7 Angle (STA), S2–L5 Angle (SLA), Spinal–Pelvic Angle (SPA), Cervical Tilt (CT), Dorsal Angle (DA), Lumbar Angle (LA), Sagittal Vertical Axis (SVA), Heel–S2–Nasion (HSN), Heel–S2–C7 (HSC), Heel–S2–T7 (HST), Shoulder–Elbow Angle (SEA), Elbow Flexion (EF) and Wrist–SIAS Offset (WSO). These additional kinematic parameters are described and illustrated in Figure 1B,C. All kinematic parameters were normalized to the gait cycle from 0 to 100 represented on the *y*-axis; on the *x*-axis, a range of 50 degrees or cm was used as the scale of the graph in order to include all Excursion Ranges, standardize the size of the kinematic curves and better underline the different inter-subject repeatability. Data obtained for each parameter from each trial were averaged with Smart Analyzer to obtain an output kinematic parameter representing the trend of each parameter during the gait cycle (kinematic curve).

### 2.4. Statistical Analysis

Data exported from BTS were imported into Matlab [22] software for further processing. No significant differences between the left and right sides for neither spatial–temporal nor kinematic parameters were found via non-parametric Wilcoxon signed-rank tests; therefore, only left gait cycles were considered for subsequent statistical analyses. Mean values and Standard Deviations (SDs) across all subjects were calculated for the spatial–temporal parameters. Inter-subject variability in the latter was assessed with the Coefficient of Variation (CV), that is, the ratio between the SD and the mean value of each parameter. Afterwards, kinematic curves were processed with an “ad hoc” script in Matlab in order to obtain the following new additional inter-subject kinematic values: Excursion Range, Absolute Excursion Range, Average and SD Average (Figure 2, graphs δ and ɣ). “Excursion Range” is the distance between the minimum and maximum of each kinematic curve during the gait cycle for each subject (the mean and SD of this value were calculated across all subjects for each new parameter and are reported in Table 3). “Absolute Excursion Range” is the distance between the Absolute Maximum and Minimum of the kinematic curves across all subjects for each parameter. “Average” is the mean trend of the kinematic curves across all subjects for each parameter during the gait cycle (it is the kinematic curve graphically represented in Figure 2); the mean and SD of the Average of each parameter were calculated. Eventually, “SD Average” was computed as the Average of the SD during the gait cycle across trials, which were pooled as follows: inter-trial, across 14 groups (1 examiner * 14 subjects * 1 session). The latter value, determined according to [23,24], had the same unit of measure of the referenced parameter and characterized the inter-subject repeatability across trials of each kinematic parameter. The intra-subject repeatability of the kinematic parameters was assessed with the Coefficient of Variation (CV), that is, the ratio between the SD and the mean value of the reciprocal of the Excursion Range area (area within minimum and maximum of the Excursion Range) of each kinematic parameter.

## 3. Results

The mean, SD and median of the baseline characteristics of healthy participants are shown in Table 1. The results of the spatial–temporal parameters are reported in Table 2; they highlighted very small SD and Excursion Range values for all spatial-temporal parameters (low inter-subject variability, CV < 7%) except for the gait speed, the double-support phase and the step width, which showed slightly larger inter-subject variability (CV~8–15%).

**Table 2 jfmk-07-00057-t002:** Mean and Standard Deviation (with minimum and maximum values of range) and Coefficient of Variation for each spatial–temporal parameter in healthy subjects. Standard Deviation (SD), Coefficient of Variation (CV).

Spatial–Temporal Parameters	Mean ± SD (Min, Max)	CV (%)
Cycle Duration (s)	1.15 ± 0.09 (1.04; 1.34)	7.8%
Cadence (step/min)	104.8 ± 8.00 (89.55; 113.2)	7.6%
Gait Speed (m/s)	1.06 ± 0.13 (0.9; 1.3)	12.2%
Stance Phase (%)	64.36 ± 1.54 (61.41; 66.31)	2.4%
Swing Phase (%)	35.53 ± 1.64 (33.09; 38.59)	4.6%
Double-Support Phase (%)	14.26 ± 2.06 (10.6; 17.09)	14.4%
Stride Length (m)	0.610 ± 0.060 (0.520; 0.740)	9.8%
Step Width (m)	0.090 ± 0.002 (0.050; 0.110)	0.2%

The means and SDs of Average and Excursion Range values, Absolute Excursion Range with Absolute Maximum and Minimum, intra-subject CV and SD Average for each new kinematic parameter are respectively reported in Table 3.

**Table 3 jfmk-07-00057-t003:** Mean value, Mean Excursion Range, SD Average, Absolute Excursion Range (absolute inter-subject min and max degree of curves for each group), Coefficient of Variation (CV) of intra-subject repeatability and inter-subject repeatability. Dorsal Angle (DA), Lumbar Angle (LA), Elbow Flexion (EF), Shoulder–Elbow Angle (SEA), Sagittal Vertical Axis (SVA), Wrist–ASIS Offset (WSO), C7–Nasion Angle (CNA), T7–Nasion Angle (TNA), S2–Nasion Angle (SNA), S2–C7 Angle (SCA), S2–T7 Angle (STA), S2–L5 Angle (SLA), He–S2 Angle (HSA), He–S2–Ns Angle (HSN), He–S2–C7 Angle (HSC), He–S2–T7 Angle (HST), Spinal–Pelvic Angle (SPA). Maximum (Max), Minimum (Min), Standard Deviation (SD), Coefficient of Variation (CV). The parameters with CV < 50% (high intra-subject repeatability) and with SD Average ≤ 1 (high inter-subject repeatability) are highlighted in bold.

New Kinematic DB-Total Parameters	Average Mean ± SD	Excursion Range Mean ± SD	Absolute Excursion Range (Absolute Max; Min)	CV (%) Intra-Subject Repeatability	SD Average Inter-Subject Repeatability
HSA (Deg)	−0.17 ± 1.50	38.8 ± 7.1	48.5 (23.9; −24.6)	**40.9%**	**0.8**
HST (Deg)	183.61 ±1.87	41.5 ± 7.9	52.5 (209.4; 156.9)	**34.1%**	**1.0**
HSC (Deg)	173.01 ± 1.70	35.6 ± 13.2	53.0 (200.0; 147.0)	**28.5%**	**0.9**
HSN (Deg)	155.66 ± 2.20	36.8 ± 13.9	52.1 (184.2; 132.1)	**48.7%**	1.3
SNA(Deg)	24.19 ± 2.52	2.8 ± 0.7	13.1 (31.1; 18)	54.5%	1.4
TNA (Deg)	42.65 ±2.44	3.6 ± 1.3	13 (49.6; 36.6)	75.8%	1.9
CNA (Deg)	61.37 ± 5.96	4.1 ± 1.8	20 (71.4; 50.4)	62.2%	3.1
CT (Deg)	−47.05 ± 15.47	5.1 ± 2.2	49.6 (−24.9; −74.5)	59.7%	3.6
SLA (Deg)	14.08 ± 5.96	4.2 ± 1.6	25.3 (26.9; 1.6)	**36.7%**	1.3
STA (Deg)	−3.78 ± 2.37	2.7 ± 0.8	12.3 (3.7; −8.6)	**30.5%**	**1.0**
SCA (Deg)	6.82 ± 1.84	2.5 ± 0.7	9.6 (11.8; 2.2)	**35.9%**	**1.0**
SVA (cm)	5.6 ± 1.4	2.7 ± 1.3	8.0 (10.0; 2.0)	**39.9%**	**0.9**
SPA (Deg)	127.99 ± 5.65	3.5 ± 1.5	18.8 (136.2; 117.4)	52.3%	1.8
LA (Deg)	196.72 ± 7.02	5.2 ± 2.7	33.7 (220.6; 186.9)	**31.5%**	1.1
DA (Deg)	149.45 ± 5.38	1.7 ± 0.5	18.4 (157.5; 139.1)	**30.5%**	**0.7**
WSO (cm)	−2.1 ± 3.7	34.5 ± 17.6	50 (19.0; −32.0)	**36.9%**	2.3
SEA (Deg)	7.33 ± 2.99	25.9 ± 8.5	39.5 (29.2; −10.3)	**22.6%**	2.1
EF (Deg)	146.80 ± 6.22	21.3 ± 7.8	35.3 (167.2; 131.9)	**32.6%**	1.8

Moreover, the Average of each DB-Total parameter is represented as a kinematic curve in Figure 2. Low intra-subject variability (CV < 50% = high repeatability) of the new kinematic parameters was found for most parameters (HSA, SCA, STA, SLA, DA, LA, SVA, HSN, HSC, HST, SEA, EF and WSO), while higher intra-subject variability (CV > 50% = lower repeatability) was detected for the other ones (SPA, CNA, TNA, SNA and CT, as shown in Table 3). Low inter-subject variability (SD Average ≤ 1 = high repeatability) was reported for many DB-Total parameters (HSA, SCA, STA, DA, SVA, HSC and HST), while slightly larger variability (1 < SD Average < 2) was found for TNA, SNA, SPA, SLA, LA, HSN and EF; the highest inter-subject variability (>2) was shown for CNA, CT, SEA and WSO, as shown numerically in Table 3 and graphically in Figure 2 (observing the width of the SD between kinematic curves).

**Figure 2 jfmk-07-00057-f002:**
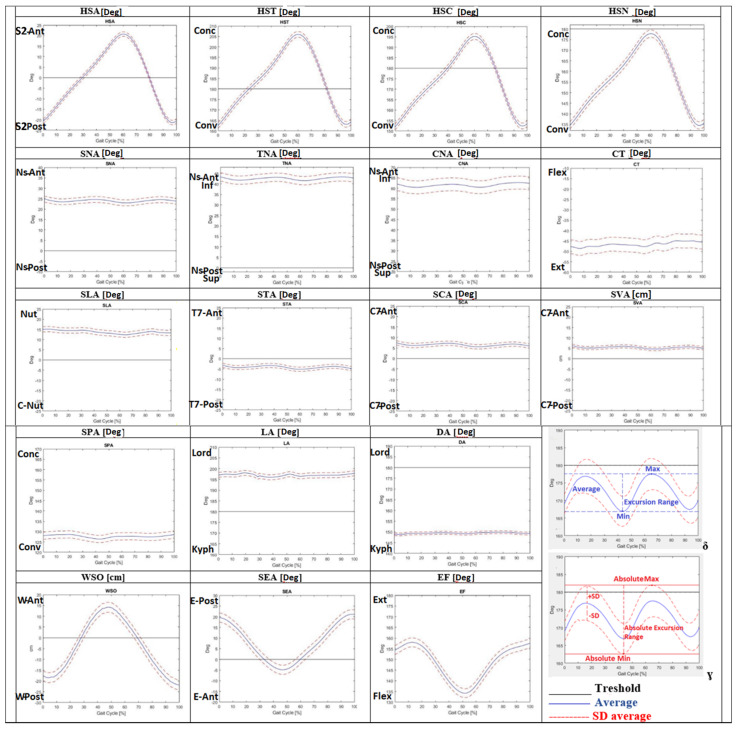
Average of kinematic curves during the gait cycle for eighteen new sagittal DB-Total [14] parameters. Graphs δ and ɣ explain the new kinematic DB-Total values: Average, Excursion Range, Absolute Excursion Range, Minimum (Min), Maximum (Max), Standard Deviation (SD), and Absolute Minimum (Min) and Maximum (Max). The scale range of the graph was set to 50 (degrees or cm) in order to highlight different inter-subject variability. Dorsal Angle (DA), Lumbar Angle (LA), Elbow Flexion (EF), Shoulder–Elbow Angle (SEA), Sagittal Vertical Axis (SVA), Wrist–ASIS Offset (WSO), C7–Nasion Angle (CNA), T7–Nasion Angle (TNA), S2–Nasion Angle (SNA), S2–C7 Angle (SCA), S2–T7 Angle (STA), S2–L5 Angle (SLA), He–S2 Angle (HAS), He–S2–Ns Angle (HSN), He–S2–C7 Angle (HSC), He–S2–T7 Angle (HST), Spinal–Pelvic Angle (SPA), nasion (Ns), wrist (W), elbow (E), posterior (Post), anterior (Ant), dorsiflexion (Dors), plantiflexion (Plant), flexion (Flex), extension (Ext), adduction (Add), abduction (Abd), kyphosis (Kyph), lordosis (Lord), nutation (Nut), counter nutation (CNut), concavity (Conc), convexity (Conv).

## 4. Discussion

The present study aimed to characterize the sagittal spinal and whole-body posture of healthy subjects during walking. The use of the DB-Total protocol [14] allowed us to measure additional whole-body kinematic parameters during walking with respect to traditional stereophotogrammetric protocols [7,8,9,10,11], which usually consider the trunk as a single rigid segment, without information on the kinematic changes within the spine about the head and the upper limbs. A few studies evaluated multi-segmental trunk and whole-body angles considering a standing posture [12,13,14] or during walking [15,16,17,18,19,20,21]. Our study analyzed the DB-Total parameters (characterizing sagittal spinal and whole-body kinematics and their intra-subject and inter-subject repeatability) and calculated new additional values (Excursion Range, Absolute Excursion Range, Average and SD Average).

The results of the spatial–temporal parameters (Table 2) highlighted very small SD and low inter-subject variability (CV < 7%) except for the gait speed, the double-support phase and the step width (CV~8–15%), underlining the high repeatability of all s–t parameters and the slightly larger variability of the latter; these findings indicated a good homogeneity of walking across all healthy subjects, which makes the statistical analysis of kinematic parameters more reliable.

Regarding the new DB-Total parameters, the low intra-subject variability (CV < 50%) of most of them (HSA, HSN, HSC, HST, STA, SCA, SVA, SLA, DA, LA, SEA, EF and WSO) showed high intra-subject repeatability of their kinematic curve during the gait cycle. Instead, the higher intra-subject variability (CV > 50%) of SPA, CNA, TNA, SNA and CT revealed low repeatability, highlighting a less consistent pattern for the parameters related to the cervical spine–head, probably for the greater mobility of this body district (Table 3). Low inter-subject variability (SD Average < 1) was reported for HSA, HSC, HST, SCA, SVA, STA and DA and slightly larger (<2) for HSN, TNA, SNA, SPA, SLA, LA and EF, while higher inter-subject variability (>2) for CNA, CT, SEA and WSO was found, revealing the high inter-subject repeatability of the former parameters related to more fixed body regions (pelvis–dorsal spine) with respect to more mobile ones (cervical spine–head–upper limbs) (Table 3, Figure 2). Another biomechanical explanation may relate to the need of maintaining the heaviest and most rigid body districts (pelvis and rib cage) inside the support area between the two feet; therefore, the heel–sacrum–T7/C7 segments should be aligned, while more mobile ones (lower limbs, lumbar and cervical spine) can adapt to movement. These results are in agreement with those in [18], which presented great variability in the cervicothoracic junction (SD = 13.7°) of healthy adolescents and young adults, confirming a larger mobility of this anatomical region with respect to the others. Moreover, high intra-subject but low inter-subject repeatability was found for SEA, EF and WSO (Table 3, Figure 2), revealing an intra-subject homogeneous kinematic pattern of the upper limbs that varied across the subjects because of different postural attitudes. Therefore, these parameters should be used with caution in any comparison with larger healthy or pathological populations to identify abnormalities during walking. 

The results of our study defined the kinematic curves and normal ranges of the new DB-Total parameters that analyzed, bottom-up, the correlations among the positions of the heels, pelvis, multi-segmental trunk, head and upper limbs during the different phases of the gait cycle. The Average of the Heel–S2 Angle showed the anterior position of the sacrum with respect to the heel from the mid-stance to mid-swing phases and then a posterior one from the mid-swing to the mid-stance (Figure 2). A similar kinematic trend was found for the Average of HST, HSC and HSN (Figure 2). These angles investigated the sagittal alignment among Heel, S2 and, respectively, T7, C7 or Nasion, representing “sagittal whole-body arches” (with S2 as the vertex), and showed a peak of concavity in the phases of pre-/initial swing and of convexity in the terminal swing (Figure 2 and Figure 3). In the latter phase, therefore, the head–trunk and heel segments were in the most anterior position, while in the phases of pre-/initial swing, they were in the most posterior position with respect to the sacrum (Figure 3).

The analyses of the CNA, TNA and SNA Average values, always placed over the threshold line, confirmed the anterior position of the nasion with respect to C7, T7 and sacrum, respectively, during walking (Figure 2). In particular, the Average of these angles showed a similar kinematic trend, with a decrease during the loading response and the pre-swing, and an increase in the terminal stance and the mid-swing (Figure 2 and Figure 3). These trends highlighted the need for trunk–head alignment before the single-support phase and for head anteriorization during the single-support phase in order to promote, respectively, the stability and forward propulsion of the body during walking. These findings were confirmed by the similar trend of the STA, SCA and SVA Average values (Figure 2); the former was found to be below the threshold line, showing a mild posterior position of T7 with respect to the sacrum, while the latter ones were always represented over the threshold lines, underlining the anterior position of C7 with respect to the sacrum during the entire gait cycle (Figure 2 and Figure 3). Our results of the dynamic SVA are in agreement with previous studies [19,20,21], where a positive mean value and very small ROM (as Excursion Range) were found. Another study [18] reported a greater mean value and ROM than those in our findings, probably because of the younger age of the sample, which can show a more dynamic and variable kinematic pattern. The analysis of the Cervical Tilt angle, which expresses the relation between the head (horizontal plane of the gaze) and the trunk (plane of upper thoracic outlet), was always found to be below the threshold line, underlining the relative extension of the head with respect to the upper-trunk outlet during the whole gait cycle. Regarding the sagittal spinal angles, the Dorsal Angle showed a constant Average of about 150° (kyphosis angle with a small Excursion Range of 2°) and the Lumbar Angle one of about 197° (lordosis angle with a slightly larger Excursion Range of 5°). The Dorsal Angle Excursion Range being shorter than the Lumbar one during walking depended on the more rigid and stable structure of the dorsal spine; these results are in agreement with those in [18,20,21] in regard to the mean value and Excursion Range. The analysis of the S2–L5 Angle revealed an Average over the threshold line and a mean value of 14.08, indicating a trend of horizontalization and nutation of the Sacrum–L5 plate during the gait cycle, with a mild increase in the initial contact–mid-stance and a decrease in the initial swing phase (Figure 2), which was very similar to the sagittal pelvic pattern because of the close anatomical relationships. Another investigated angle was the Spinal–Pelvic Angle, which analyzed the relation between C7–S2 and the center of the femoral head. This angle is an intrinsic parameter of balance, as well as the corresponding radiological spino-sacral angle [25]; its Average under the threshold line and mean value of 128° underlined a convexity with an almost constant kinematic pattern (Figure 2 and Figure 3). Finally, the quantitative evaluation of upper-limb kinematics during the gait cycle showed an Average of the Shoulder–Elbow Angle with a positive value (meaning a posterior position of the elbow with respect to the shoulder) and a negative one during the last part of the terminal stance and pre-swing (40–60%, with an anterior position of the elbow with respect to the shoulder). The kinematic pattern was characterized by two peaks, a positive one in the terminal-swing/initial-contact phases, which indicated the maximum posteriorization of the elbow with respect to the shoulder, and a negative one in the pre-swing, which indicated the maximum anteriorization of the elbow (Figure 2 and Figure 3). A similar trend was found for the Elbow Flexion angle, showing an Average under the threshold line (<180°) with flexion during all the gait cycle and the maximum peak in the pre-swing (50–60%) (Figure 2 and Figure 3). Eventually, the analysis of the Wrist–ASIS Offset Average showed a posterior position of the wrist with respect to the ASIS during the gait cycle except for the terminal stance, and the phases of pre- and initial swing (30–70%), with a negative peak (maximum posteriorization of the wrist with respect to the ASIS) in the initial contact and a positive peak (maximum anteriorization of the wrist) in the pre-swing phase. 

*Limitations:* There are some limitations associated with the present study, such as the relatively small sample size and the heterogeneity of the recruited group with respect to the age, weight and height. However, the use of 3D motion analysis (gold-standard instrumental examination for static and dynamic postural evaluation), the accuracy of the DB-Total protocol and the assignment of markers placement to a single expert physician make our results reliable. The normal dataset may be used for comparison with a pathological population, above all for the most repeatable kinematic parameters. Nevertheless, future studies on a larger population would be needed in order to verify inter-subject variability in relation to different features (age, weight, height, etc.) of the healthy population.

## 5. Conclusions

The aim of the present study was to assess the sagittal posture of an adult healthy population using 3D motion analysis. The use of the DB-Total marker set let us better investigate spinal and multi-segmental body kinematics, defining new additional parameters, and their intra-subject and inter-subject repeatability.

Despite the aforementioned limitations of the study, our results revealed typical spinal and whole-body kinematic patterns in the healthy population that may explain the biomechanical total-body strategies for maintaining balance during walking. The use of DB-Total parameters and the normal dataset might help to diagnose and better understand whole-body kinematic deviations in an adult pathological population.

## Figures and Tables

**Figure 1 jfmk-07-00057-f001:**
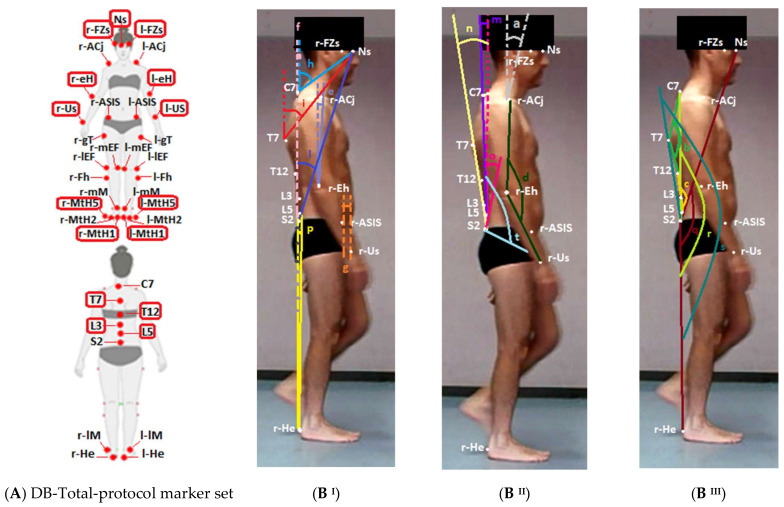
(**A**) DB-Total marker set protocol. (**B**,**C**) Graphic representation and definition of eighteen sagittal whole-body parameters. (Figures (**B**) show an adult healthy subject during walking at 20% of gait cycle). Additional DB-Total markers were circled in red with respect to Helen Hayes Medial Markers set ones [11] (**A**). 37 reflective markers were placed on nasion (Ns), frontozygomatic suture (FZs), spinous apophysis of C7 -T7 -T12 -L3 -L5 -S2, acromioclavicular joint (ACj), epicondylus humeri (eH), ulnar styloid (Us), anterior superior iliac spine (ASIS), greater trochanter (gT), medial (mEF) and lateral (lEF) epicondylus femoris, fibular head (Fh), medial (mM) and lateral (lM) malleoli, I°–III° and V° metatarsal heads (MtH) and heel (He). UM (unit of measure). T (threshold: the line that defines the passage from one direction of movement to the opposite for each kinematic parameter, as shown on kinematic graphs in Figure 2). Vertical axes are represented as dashed lines.

**Figure 3 jfmk-07-00057-f003:**
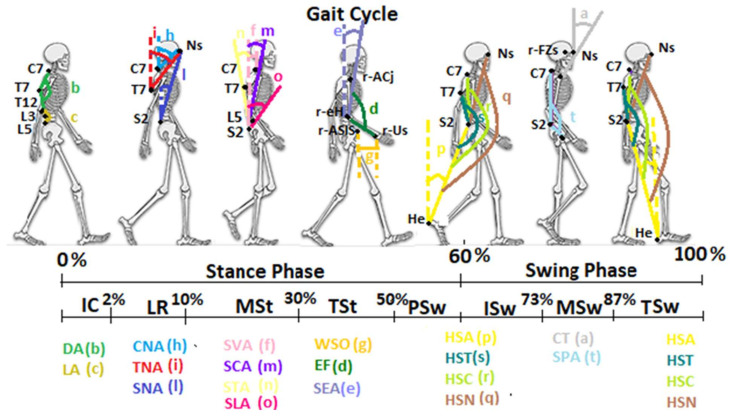
Graphic representation of the new sagittal DB-Total parameters during the different phases of the gait cycle. Initial contact (IC), loading response (LR), mid-stance (MSt), terminal stance (TSt), pre-swing (PSw), initial swing (ISw), mid-swing (MSw), terminal swing (TSw), nasion (Ns), heel (He), sacrum (S2). The acronyms and the colors of the parameters are the same as those mentioned in Figure 1. Vertical axes are represented as dashed lines.

**Table 1 jfmk-07-00057-t001:** Mean and Standard Deviation for baseline features of healthy subjects. Standard Deviation (SD), years (Y), male (M), female (F), Body Mass Index (BMI).

Baseline Features	
Age (y) mean ± SD (median)	46.7 ± 14.9 (45)
Gender (number, F–M)	7 F–7 M
Height (m) (mean ± SD)	1.71 ± 0.10
Weight (kg) (mean ± SD)	72.3 ± 21.6
BMI (kg/m^2^) (mean ± SD)	23.44 ± 2.99

## Data Availability

The data that support the findings of this study are available from the corresponding author upon reasonable request.

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
