# Peer review of "Kinematic Evaluation of the Sagittal Posture during Walking in Healthy Subjects by 3D Motion Analysis Using DB-Total Protocol"

_jfmk, 2022, doi:10.3390/jfmk7030057_

Round 1

Reviewer 1 Report

1 - In the abstract, I suggested that the results and conclusions were more assertive. In the present format, they are repetitions of content, referring to practically the same thing. They must respond objectively to research questions.

2 - The introduction can be improved, referring to some of the limitations that exist with the methodologies commonly used. This may justify the need for more scientific investment.

3 - In methods, I suggested the description of the type of study, and in the exclusion criteria, postural disorders of the spine were not considered.

I consider important to know who makes the placement of the markers. What is the profession and what is the experience, as well as scientific recognition.

I suggested a review of the discussion, making it more didactic and promoting the comparison of this research with other existing ones. It was important to discuss these specificities in the sagittal posture and what their possible impact / usefulness in the medical clinic.

4 - Conclusion, as mentioned by the authors, the small sample, the morphological heterogeneity and the non-reference to the exclusion of sagittal disorders make the conclusions poorly supported (or to appear with some caution).

Author Response

Response to Reviewers

Comments and Suggestions for Authors

Reviewer 1

1 - In the abstract, I suggested that the results and conclusions were more assertive. In the present format, they are repetitions of content, referring to practically the same thing. They must respond objectively to research questions.

The authors thank the reviewer for the suggestion and added a short phrase to explain better the results. Nevertheless, we have tried to summarize the results by showing the kinematic curves of DB-Total parameters and underlining their high intra- and inter-subject repeatability, within the limit of the abstract of 200 words

2 - The introduction can be improved, referring to some of the limitations that exist with the methodologies commonly used. This may justify the need for more scientific investment.

The text has been modified, underlining the limitation of other study about the absence of whole-body parameters that, instead, were considered in our study.

3 - In methods, I suggested the description of the type of study, and in the exclusion criteria, postural disorders of the spine were not considered.

Thanks for the correct suggestions. The text has been revised accordingly.

I consider important to know who makes the placement of the markers. What is the profession and what is the experience, as well as scientific recognition.

The text has been revised accordingly.

I suggested a review of the discussion, making it more didactic and promoting the comparison of this research with other existing ones. It was important to discuss these specificities in the sagittal posture and what their possible impact / usefulness in the medical clinic.

The authors may understand the reviewer’s suggestion, but disagree on the lack of didacticity of the discussions. In fact, we tried to explain each parameter considering kinematic curve and additional values, the clinic meaning of intra- and inter-subject repeatability, and suggesting the biomechanical interpretation.

4 - Conclusion, as mentioned by the authors, the small sample, the morphological heterogeneity and the non-reference to the exclusion of sagittal disorders make the conclusions poorly supported (or to appear with some caution).

The text has been revised accordingly.

Author Response

Response to Reviewers

Comments and Suggestions for Authors

Reviewer 2

The following observations/comments are drawn out of this work, and required to address these comments by authors before accepting for the publication.

  1. All the abbreviations are needs to be inserted at the start or end of the manuscript. Example, Coefficient of variation (CV) and inter-subjects Standard Deviation Average (SD-Average), Postural Control System (PCS).

The text has been revised accordingly.

  1. Limitations needs to be subchapter in discussion.

The text has been revised accordingly.

  1. All this work considered under what speed of walking, is itself selected speed or particular speed. If so how you can justify this frame works helps in other activity.

Each subject performed three consecutive trials at a self-selected normal paced speed. All spatial temporal parameters showed a low coefficient of variation, gait speed too (12.2%), indicating a good repeatability among recruited subjects.

  1. Have you compared the results with exoskeletal assisted walking devices, because this gives a clear idea that how this frame work helps to evaluate the exoskeletal assisted walking? Usually after SCI people generally depends on these assisted walking. Example like wandercraft, kneego

No, this study didn’t compare the results with exoskeletal assisted walking devices. To the best of our knowledge, the studies that used exoskeletal assisted walking devices always considered sagittal kinematics of trunk (as single rigid segment), pelvis and lower limbs. The authors thank the authors for his very interesting suggestion, useful for future study.

  1. Is any statistical test [like paired T test] is performed to know the effect of age, sex and b/t male and female?

No statistical tests [like paired T test] are performed to know the effect of age, sex and b/t male and female, because of small sample size. Moreover, aim of our study was also to understand the intra- and inter-subject repeatability of these new parameters in general adult population, without further differences.                                                

This work is unique and interesting. This can be considered for publication after addressing the above comments.

The authors thank the reviewer for his nice comment and interest about this study.

Round 2

Reviewer 1 Report

no comments